

# Exploring phase space with Neural Importance Sampling

**Enrico Bothmann, Timo Janßen, Max Knobbe, Tobias Schmale and Steffen Schumann**

Institut für Theoretische Physik, Georg-August-Universität Göttingen,
D-37077 Göttingen, Germany

## Abstract

We present a novel approach for the integration of scattering cross sections and the generation of partonic event samples in high-energy physics. We propose an importance sampling technique capable of overcoming typical deficiencies of existing approaches by incorporating neural networks. The method guarantees full phase space coverage and the exact reproduction of the desired target distribution, in our case given by the squared transition matrix element. We study the performance of the algorithm for a few representative examples, including top-quark pair production and gluon scattering into three- and four-gluon final states.

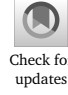
# 1 Introduction

An important deliverable in high-energy particle physics are quantitative predictions for the outcome of collider experiments. This includes total and differential production rates in the framework of the Standard Model or hypothetical New Physics scenarios. To allow for a direct comparison with experimental data, multi-purpose event generators such as PYTHIA [1], HERWIG [2] or SHERPA [3, 4] proved to be vital tools. Starting from the evaluation of partonic hard-scattering cross sections they accomplish a fully differential and exclusive simulation of individual scattering events by invoking parton shower simulations, particle decays, models for the parton-to-hadron transition and, in case of composite colliding entities such as protons, multiple interactions per collision. See [5] for a recent review of Monte Carlo event generators.

In contemporary Standard Model analyses as well as searches for New Physics, hard-scattering processes featuring a rather high multiplicity of final-state particles are of enormous phenomenological relevance. This includes in particular signatures with multiple hard jets or a number of intermediate resonances that decay further on. Illustrative examples are the production of V + jets final states or top-quark pair production in association with a boson $V = \gamma, H, Z^0, W^\pm$ in proton-proton collisions at the LHC. Such cutting-edge channels require the efficient evaluation of the corresponding partonic scattering matrix elements, featuring up to 8 final-state particles, with easily thousands of Feynman diagrams contributing. This clearly goes well beyond the traditional realm of multi-purpose generators, such that specialised tools for this computationally very intense task have emerged over time, known as matrix element generators or parton-level event generators. These tools largely automate the generation and evaluation of almost arbitrary scattering matrix elements. At tree-level this includes tools such as AMEGIC [6], COMIX [7], MADGRAPH [8, 9] or WHIZARD [10]. For one-loop matrix elements widely-used examples are MADGRAPH5_AMC@NLO [9], OPENLOOPS [11], POWHEGBOX [12] or RECOLA [13, 14]. Equipped with a phase space generator these tools can be used to compile partonic cross section evaluations, to calculate decay widths and to probabilistically generate partonic events. When incorporated into or interfaced to a multi-purpose event generator they provide the momentum-space partonic scattering events that seed the evolution to fully exclusive particle-level final states.

State-of-the-art matrix element generators use adaptive Monte Carlo techniques for generating phase space points with a distribution that reasonably approximates the target distribution, such that event weight fluctuations are reduced. Samples of unit-weight events can then be generated with a distribution given by the actual target function by applying a simple hit-or-miss algorithm. However, nowadays matrix element generators are often limited by the performance of their phase space sampler. An insufficient mapping of the target distribution results in significant fluctuations of the event weights and correspondingly a large number of target-function evaluations are needed when generating unit-weight events.

Typically the sampling performance deteriorates significantly with the phase space dimensionality, i.e. particle multiplicity [15], and the complexity of the integrand. In particular the appearance of intermediate resonances, regularised singularities or quantum-interference effects complicate the situation. Further limitations arise from non-trivial kinematical cuts that the integrator can not address, i.e. adapt to.

Compared to the efforts that went into the development of improved scattering-amplitude construction algorithms, the field of phase space sampling has seen rather little conceptual developments. For some recent works see [16–21]. Besides the matrix element generator implementations, there are public libraries like CUBA [22] (implementing the VEGAS, DIVONNE, SUAVE, CUHRE algorithms) or FOAM [23, 24] that are widely used. There have been some efforts to employ Markov Chain techniques for phase space sampling, cf. [25, 26]. However, very recently there has been significant interest to employ modern machine-learning techniques

to the problem of phase space sampling in particle physics, cf. [27–31]. The tremendous advances in the field of machine learning, driven from very different applications such as image generation or light-transport simulation, also fuel the work we present here.

The paper is organised as follows. In Sec. 2 we review the basics of Monte Carlo integration and phase space sampling techniques as used in high-energy event generators, and discuss potential pitfalls when extending or replacing these methods using neural networks. In Sec. 3 we present our novel sampler that inherits all the properties of an importance sampler, but with the phase space mapping optimised through bijective maps, so-called coupling layers [32]. These are adjusted by training neural networks, which has originally been proposed in [32]. Our work is in principle an application of 'Neural Importance Sampling' [33] as we employ the 'polynomial coupling layers' introduced therein, although we want to point out that the usage of coupling layers for importance sampling has also been studied in [34]. In Sec. 4 we discuss benchmark applications of our method from high-energy physics, including top-quark pair production and gluon scattering into three- and four-gluon final states. Conclusions and a brief outlook are presented in Sec. 5.

An independent study of applying Neural Importance Sampling to high-dimensional integration problems is simultaneously presented in [35], and a follow-up application of this approach to HEP processes appeared in [36].

## 2 Phase space sampling: existing approaches

To set the scene we start out with a brief review of the basics of Monte Carlo integration and event sampling. For ease of having a clean nomenclature we consider a simple positive-definite target distribution $f : \Omega \subset \mathbb{R}^d \to [0, \infty)$ defined over the unit hypercube, i.e. $\Omega = [0, 1]^d$. In our use case hypercube points $u_i \in \Omega$ are mapped onto a set of final-state four-momenta $\{p_i\}$, the corresponding Jacobian is considered part of the integrand $f(u_i)$. The phase space dimensionality $d$ is set by the number of final-state particles $n$, i.e. $d = 3n-4$. We thereby implement on-shell constraints for all external particles and total four-momentum conservation. There are two standard tasks that we wish to address in what follows, the probabilistic generation of phase space points according to the target distribution $f$ and the evaluation of integrals over $f$.

The Monte Carlo estimate of the integral over the unit hypercube

$$I = \int_\Omega f(u') \, du' \tag{1}$$

is given by

$$I \approx E_N = \frac{1}{N} \sum_{i=1}^{N} f(u_i) = \langle f \rangle, \tag{2}$$

where we assumed uniformly distributed random variables $u_i \in \Omega$. The corresponding standard deviation, when assuming large $N$, is given by

$$\sigma_N(f) = \sqrt{\frac{V_N(f)}{N}} = \sqrt{\frac{\langle f^2 \rangle - \langle f \rangle^2}{N}}, \tag{3}$$

with $V_N$ the corresponding variance.

Interpreting the random points $u_i$ as individual *events*, we call $f(u_i)$ the corresponding *event weight* $w_i$, such that the integral is estimated by the average event weight $\langle w \rangle_N$. When asked to generate $N$ unit-weight events according to the distribution $f(u)$, a simple hit-or-miss

algorithm can be employed to convert a sample of weighted events into a set of unweighted events. The corresponding unweighting efficiency is given by

$$\epsilon_{\text{uw}} := \frac{\langle w \rangle_N}{w_{\text{max}}},$$

(4)

with $w_{\text{max}}$ the (numerically pre-determined) maximal event weight in the integration region. An efficient integrator, i.e. sampler, foremost aims for a reduction of the variance $V_N$ that will typically result in an increased unweighting efficiency $\epsilon_{\text{uw}}$. Even though these two figures of merit are interrelated, they provide complementary means for the optimisation of a sampler, i.e. reducing the variance does not necessarily yield an improved unweighting efficiency. In the next section we will discuss established methods that achieve a variance reduction.

We close this introductory section by specifying requirements we impose on our improved cross section integration and parton-level event generation algorithm:

(i) The samples produced by the algorithm should converge to the true target distribution everywhere in phase space.[1]

(ii) We demand that the full physical phase space is to be covered for the limit $N \to \infty$. This should be guaranteed, even if potential training samples only feature finite statistics and thus provide no full coverage of the available phase space volume.

(iii) The method should be general, lending itself to automation. By that we wish the algorithm to be self-adaptive to new integrands, without the need of manual intervention.

(iv) The method should be capable of producing samples of uncorrelated events.[2]

As discussed in the following, these conditions are naturally fulfilled by traditional sampling algorithms used in high-energy physics, such as importance and stratified sampling. However, this is not necessarily true for some of the recently proposed samplers based on neural networks as discussed in Sec. 2.3. In Sec. 3 we will present our novel algorithm employing neural-network techniques, that indeed fulfils all the above criteria.

## 2.1 Importance Sampling

As can be seen from Eq. (3) the standard deviation of a Monte Carlo integral estimate scales as $1/\sqrt{N}$, independent of the dimensionality of the problem. However, besides the sample size, the variance of the integrand over the integration region determines the quality of the integral estimate and in turn the unweighting efficiency $\epsilon_{\text{uw}}$. In particular for strongly structured, possibly multi-modal target distributions it is therefore vital to introduce specific variance-reduction techniques to obtain more accurate integral estimates for a given sample size.

To this end a suitable variable transformation can be utilised, i.e. producing phase space points with a positive definite non-uniform distribution function $G(u) : \Omega \mapsto \Omega$, such that

$$I = \int_{\Omega} \frac{f(u')}{g(u')} g(u') \, du' = \int_{\Omega} \frac{f(u')}{g(u')} \, dG(u') = \left\langle \frac{f}{g} \right\rangle,$$

(5)

with $g(u) : \Omega \mapsto \mathbb{R}$. The relevant variance is thus $V(f/g)$. Hence it can be significantly reduced by picking $g(u)$ similar in shape to $f(u)$. Obviously, the optimal choice would be

$$G(u) = \int_{\Omega} f(u') \, du', \quad \text{i.e.} \quad g(u) = f(u).$$

(6)

---

[1]On the same basis, in [37] it has been cautioned against the usage of Generative Adversarial Networks to extrapolate from finite-statistics training data to large-scale event samples for physics analyses.

[2]This limits the use of algorithms based on Markov Chain samplers, cf. [26].

However, this presupposes the solution of the actual integration problem.

We often have to deal with multimodal targets. In that case it can be very hard to find a density that allows for efficient importance sampling. To simplify the task, we can use a mixture distribution

$$g(x) = \sum_{j=1}^{N_c} \alpha_j g_j(x), \tag{7}$$

where the $g_j$ are distributions and $\sum_{j=1}^{N_c} \alpha_j = 1$. Using a mixture distribution for importance sampling is known as *multi-channel importance sampling*. The corresponding integral estimate is given by

$$I \approx E_N = \frac{1}{N} \sum_{i=1}^{N} \frac{f(x_i)}{g(x_i)} = \frac{1}{N} \sum_{i=1}^{N} \frac{f(x_i)}{\sum_{j=1}^{N_c} \alpha_j g_j(x_i)}, \tag{8}$$

where the $x_i$ are non-uniform random numbers drawn from $g(x)$.

It is easy to sample from the multi-channel distribution: for each point one channel is chosen at random according to the $\alpha_j$ and then sampled from using the inverse-transform method. It is possible to approximate a multimodal target function by using one channel per peak. The channel weights $\alpha_j$ can be optimised automatically [38].

The performance of the multi-channel method is intimately connected with the choice of channels. In practice, information about the physics problem at hand is used to choose a suitable distribution $g$. When integrating squared transition matrix elements in high-energy physics, the propagator and spin structures of a given process are known and this knowledge can be used to construct appropriate channels [39], a procedure that is fully automated in matrix-element generators.

## 2.2 VEGAS algorithm

It can be very time consuming to find a sampling distribution that results in an efficient sampler for a given target. Because of this, adaptive importance sampling algorithms have been developed. These are able to adapt automatically to a target distribution. In the following we describe the VEGAS algorithm [40]. It uses a product density

$$q(x) = \prod_{j=1}^{d} q_j(x_j), \tag{9}$$

where each $q_j$ is a piecewise-constant function. The idea is to split the range $[0, 1)$ into $N_j$ bins $I_{j,l} = [x_{j,l-1}, x_{j,l})$, where we have defined the break points between the constant pieces as $0 = x_{j,0} < x_{j,1} < \cdots < x_{j,N_j} = 1$. The corresponding bin widths are $\Delta_{j,l} = x_{j,l} - x_{j,l-1}$ for $1 \le l \le N_j$. The functions $q_j$ are then defined by

$$q_j(x) = \frac{1}{N_j \Delta_{j,l}} \quad \text{for} \quad x_{j,l-1} \le x \le x_{j,l}. \tag{10}$$

The width of the bins can vary but per component $j$ they all have the same probability content $1/N_j$. This means that if we approximate a function with VEGAS we use many thin bins for narrow peaks and few wide bins for flat regions.

Sampling and evaluating the Jacobian for the density $q$ is straightforward. The important part is the update of the bin widths. This happens through an iterative procedure, where in each iteration we sample a number of points with the current $q$, calculate the importance weights with respect to the target $f$ and determine the new bin widths by minimising the variance for this sample. More details can be found in [40].

VEGAS is very effective for unimodal targets but has difficulties with multimodal functions if the peaks are not aligned with the coordinate axes. The density then features 'ghost peaks' which are not present in the target distribution and which can decrease the efficiency significantly.

In the simplest case, we use VEGAS to approximate a target directly in order to use the resulting density in an importance sampling scheme. However, it can be even more effective if we use it to remap the input variables (i.e. uniform random numbers) of another density, e.g. a single channel of a multi-channel distribution [41]. This amounts to adding variable transforms $\phi_j : \Omega \to \Omega$ to Eq. (7) (corresponding to the VEGAS densities $q_j$) as follows:

$$g = \sum_{j=1}^{N_c} \alpha_j (g_j \circ \phi_j^{-1}) \left| \frac{\partial \phi_j^{-1}}{\partial u} \right|,$$ (11)

with densities $g_j : \Omega \to [0, \infty)$ and hence $g : \Omega \to [0, \infty)$. As above we assume that the $\alpha_j$ sum to 1. To sample a point from this distribution we

1. randomly choose a channel according to the channel weights $\alpha_j$,

2. generate a uniform random number $u \in \Omega$,

3. use the channel-specific map $\phi_j$ to map $u$ to a non-uniform number $v$ and

4. use the inverse transform method to transform $v$ to a point $x$ according to the distribution $g_j$.

The Monte Carlo estimate of the integral is then still given by Eq. (8) but the Jacobians $\left| \frac{\partial \phi_j^{-1}}{\partial u} \right|$ of the different channels have to be taken into account.

## 2.3 Existing proposals for neural-network based sampling and possible pitfalls

A multi-layer feedforward fully connected artificial neural network (NN in the following) consists of artificial neurons arranged in several layers which are stacked on top of each other. Every neuron in a layer is connected to every neuron in the preceding layer. We distinguish the input layer, the output layer and the hidden layers in between. A single artificial neuron produces the weighted sum of its inputs and optionally adds a scalar bias. The output of the neuron is then transformed by a non-linear activation function. This means that the output $z_i^{[l]}$ of the $i$-th neuron in the $l$-th layer is given by

$$z_i^{[l]} = \vec{w}_i^{[l]T} \cdot \vec{a}^{[l-1]} + b_i^{[l]},$$ (12)

where $\vec{w}_i^{[l]}$ denotes the vector of input weights of the neuron, $b_i^{[l]}$ the respective bias and $\vec{a}^{[l-1]}$ the output of the activation function of the preceding layer. After applying the activation function $\sigma^{[l]}$ the output is given by

$$a_i^{[l]} = \sigma^{[l]}\left(z_i^{[l]}\right).$$ (13)

We assume that all hidden layers use the same activation function. The choice of output activation function is limited by the particular application. In our case it has to be a function that maps to the unit hypercube. The input layer does not use an activation function as it only passes the input variables to the neurons of the first hidden layer.

Two previous studies use this kind of NN to improve phase space sampling [27, 28]. The number of input and output neurons is there chosen equal to the number of phase space

dimensions $d$. Hence the neural network gives effectively an importance sampling mapping $g$ in the language of Sec. 2.1. The output value of the $i$th output layer neuron then gives the $i$th component of $g(u)$. Both studies described in [27, 28] use output functions that map $\mathbb{R}$ into $(0, 1)$, such that $g(u) \in (0, 1)^d$. As the hidden-layer activation function, either the hyperbolic sine, the exponential linear unit (ELU) [28] or a hyperbolic tangent [27] is used. Note that ELUs map $\mathbb{R}$ into $(-1, \infty)$ and tanh maps $\mathbb{R}$ into $(-1, 1)$, whereas sinh maps $\mathbb{R}$ into itself. The input space is $\mathbb{R}^d$ (sampled from a Gaussian distribution) in [27], and $(0, 1)^d$ in [28]. Both studies then set up different training procedures for the NN based on minimising the Kullback–Leibler (KL) divergence [42] between the NN output and the real target distribution. The details of these procedures are not important here.

What we want to point out in regards of the requirements set up in Sec. 2 is that restricting the input space to a subspace of $\mathbb{R}$ with an upper and/or lower bound will in general have the consequence that the NN map $g$ is not guaranteed to be surjective any more. The same is true if an activation function of the hidden layer maps onto a subspace of $\mathbb{R}$ with an upper and/or lower bound, such as the ELU or the tanh function. In both cases, such finite boundaries will be transformed several times, but in the end this will yield finite boundaries for the target-space coordinates, such that the support of the target distribution will be a proper subspace of the desired target space $(0, 1)^d$. Although a sufficiently long training will guarantee that the bulk of the target distribution will be within this subspace (the NN will adapt its weights to extend this subspace as required), the phase space coverage might never reach 100 %. A sample generated with such a NN will hence suffer from artificial phase space boundaries far away from the peaks of the distribution and will thus not be distributed according to the desired target distribution. Instead, it will be suppressed in the tails and enhanced in the peaks. Moreover, the artificial phase space boundaries will also yield wrong integration results. The NN structure in [28] is affected by this problem, whereas the structure in [27] is not, since it uses surjective functions throughout and the input points are given by a Gaussian distribution without a cut-off, such that the input space is given by $\mathbb{R}^d$.

To illustrate this issue, we study a simple distribution given by a 2d Gaussian centred in $(0, 1)^2$, i.e. at $(x, y) = (0.5, 0.5)$. The width of the Gaussian is set to $1/10$ of the length of the phase space edges, hence, close to the phase space boundaries the target-function values are much smaller than around the peak. We test different combinations of activation and input functions for a fully-connected NN architecture with 5 hidden layers and 64 nodes per hidden layer, always with a bounded input space of $(0, 1)^2$, as in [28]. We train the networks using the ADAM optimiser [43] with the learning rate set to $10^{-2}$. With a training data set of $N_{\text{train}} = 500\text{k}$ events, this setup yields a very poor phase space coverage for the NN regardless of the activation/output functions, namely around 25 % only:

| Input space | Activation function | Output function | Coverage (asymptote) |
| --- | --- | --- | --- |
| $(0, 1)^2$ | Sinh | Sigmoid | $0.235 \pm 0.027$ |
| $(0, 1)^2$ | ELU | Soft Clipping | $0.269 \pm 0.037$ |
| $(0, 1)^2$ | Sigmoid | Sigmoid | $0.234 \pm 0.050$ |

The coverage is estimated by the convex hull of the respective event samples, as introduced in [44]. Note that the first two rows follow the two choices discussed in [28]. The error of the asymptotic coverage is given by an average over 10 independently trained NN with different random initial weights. In Fig. 1a, we show the obtained phase space coverage as a function of the sample size $N$ for such a NN with sigmoid activation and output functions. This is compared to unweighted event samples generated from a uniform distribution and through VEGAS. In addition, the phase space coverage is also shown for a NN with surjective functions only, and with input points given by an unbounded Gaussian distribution, as in [27]. This surjective NN is guaranteed to sample the entire phase space and indeed its coverage increases

with the sample size $N$ in the same way as the uniform and VEGAS samples do, whereas the non-surjective NN shows an asymptotic behaviour in terms of the coverage. We illustrate this with three non-surjective NN setups, which are trained using $N_{\text{train}} = 250\text{k}$, 1M and 5M events, respectively. We choose these $N_{\text{train}}$ to have similar successive ratios between them, to illustrate that increasing the size of the training data successively leads to a diminishing return of investment in terms of the achieved asymptotic phase space coverage.

We study the consequence of an incomplete phase space coverage in Fig. 1b. The target distribution is averaged over bins with $x + y = \text{const.}$ (resulting in one-dimensional Gaussians) and compared with the distributions averaged in the same way given by the NN (here with $N_{\text{train}} = 500\text{k}$), by an unweighted uniform sample, by an unweighted VEGAS sample and by the averaged distribution given by the strictly surjective NN. The uniform and VEGAS samples and the one from the strictly surjective NN agree very well with the target, whereas the non-surjective NN undershoots the tails and puts too many events in the peak. We have also studied the distribution of phase space points in the two-dimensional plane, where we find that the NN is mapping the input space $(0, 1)^2$ to a slightly deformed rectangular region around the peak, which is strictly smaller than the target space.

# 3 Neural-Network assisted Importance Sampling

With the requirements stated in Sec. 2 in mind, we present our NN based approach to importance sampling. In order to be usable for multi-channel sampling, our adaptive model needs to be invertible. For this reason, we adopt the "Neural Importance Sampling" algorithm of [33]. The method of using a trainable mapping to redistribute the random numbers going into the generation of a sample is similar to how VEGAS is often used in practice. We begin this section by discussing this remapping of a distribution.

Consider a mapping $h : X \rightarrow Y, x \mapsto y$, where $x$ is distributed as $p_X(x)$. If we know $p_X(x)$ and the Jacobian determinant of $h(x)$, we can compute the PDF of $y$ using the change of variable formula:

$$p_Y(y) = p_X(x) \left| \det\left( \frac{\partial h(x)}{\partial x^T} \right) \right|^{-1} . \tag{14}$$

Using lower-upper decomposition, the cost of computing the determinant for arbitrary matrices grows as the cube of the number of dimensions and can therefore be obstructive. However, it is possible to design mappings for which the computation of the Jacobian determinant is cheap.

In [32], Dinh et al. introduce *coupling layers* which have a triangular Jacobian. As the determinant of a triangular matrix is given by the product of its diagonal terms, the computation scales linearly with the number of dimensions only. In the following, we describe the basic idea of coupling layers.

## 3.1 Coupling Layers

A coupling layer takes a $d$-dimensional input $x \in \mathbb{R}^d$. It uses a partition $\{A, B\}$ of the input dimensions $x_i$ such that $x = (x^A, x^B)$. The output $y = (y^A, y^B)$ of the coupling layer is defined as

$$
\begin{aligned}
y^A &= x^A, \\
y^B &= C(x^B; m(x^A)),
\end{aligned}
\tag{15}
$$

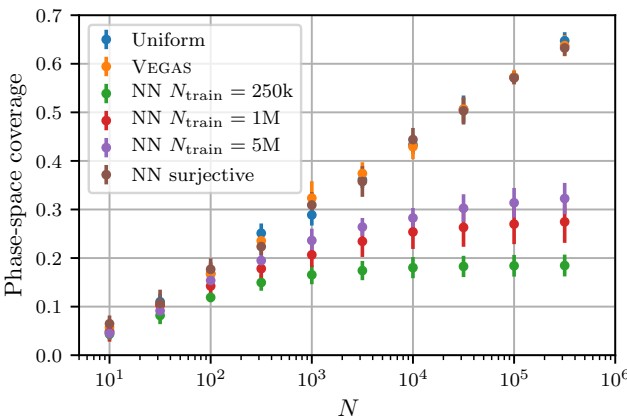

(a) The phase space coverage for different sampling methods as a function of the number of unweighted events $N$. The error bars indicate the spread over 10 statistically independent samples. For the NN, also the training has been independently repeated, each time with a new set of randomly initialised weights. For the non-surjective NN setup, the effect of using different number of training points $N_{\text{train}}$ is illustrated.

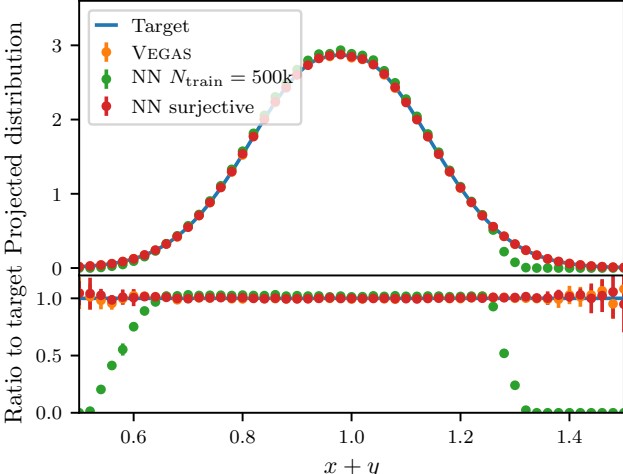

(b) The two-dimensional distribution of sampling points averaged over bins with $x + y = \text{const.}$, compared to the corresponding average of the target distribution.

Figure 1: The phase space coverage and the distribution averaged over diagonals of the two-dimensional plane for different sampling techniques, with the target distribution being a two-dimensional Gaussian centred in $(0,1)^2$. The Gaussian width is $\sigma = 0.1$. Besides the Uniform and the VEGAS samples we also show NN-generated samples. The NN architecture is described in the main text, it uses sigmoids as activation and output functions, and $N_{\text{train}} = 500$k. The input space is given by $(0,1)^2$. The "surjective" NN on the other hand only uses surjective functions and the input space is unbounded.

where the *coupling transform C* is a map that is invertible and separable, where the latter means that

$$C(x^B; m(x^A)) = \left( C_1(x_1^B; m(x^A)), \ldots, C_{|B|}(x_{|B|}^B; m(x^A)) \right)^T . \quad (16)$$

By $|B|$ we denote the cardinality of the set $B$.

According to Eq. (15) only the subset $B$ is transformed by the coupling layer, while the subset $A$ is left unchanged. Because of this, $\partial y^A / \partial (x^B)^T = 0$ and the Jacobian determinant simplifies to

$$\det\left( \frac{\partial y(x)}{\partial x^T} \right) = \prod_{i=1}^{|B|} \frac{\partial C_i(x^B; m(x^A))}{\partial (x^B)^T} . \quad (17)$$

To see this, we assume that without loss of generality we split the input dimensions in two consecutive blocks $A = [1, n]$ and $B = [n+1, d]$. In this case, the Jacobian matrix is of block form

$$\frac{\partial y(x)}{\partial x^T} = \begin{pmatrix} I_n & 0 \\ \frac{\partial C(x^B; m(x^A))}{\partial (x^A)^T} & \frac{\partial C(x^B; m(x^A))}{\partial (x^B)^T} \end{pmatrix}, \quad (18)$$

with the determinant given by Eq. (17). As the determinant does not involve the derivative $\frac{\partial m(x^A)}{\partial x^A}$, the function $m$ can be arbitrarily complex. Following [33], we represent $m$ through a NN.

A single coupling layer transforms only part of the input. To ensure that all components can be transformed, we use a layered mapping $h = h_L \circ \cdots \circ h_2 \circ h_1$, where each $h_i$ is a coupling layer. Between two layers, we exchange the roles of $A$ and $B$. For the functions $m_i$, we use one NN per layer. If $d > 3$, we need at least 3 coupling layers if we want each input component to be able to influence every output component.

Different choices for the coupling transform $C$ are possible. Additive coupling layers result in a *NICE* model [32], while affine coupling layers result in a *Real NVP* model [45]. In the following, we restrict ourselves to piecewise quadratic coupling layers, which have been proposed in [33].

### 3.2 Piecewise Quadratic Coupling Layers

We assume that our variables live in a unit hypercube: $x, y \in \Omega = [0, 1]^d$. This allows us to interpret each component $C_i$ of the coupling transform as a cumulative distribution function (CDF) in a straightforward manner. The idea is to use the output of a NN to construct unnormalised distributions $\hat{q}_i$ and get $C_i$ by integration. We normalise the distributions to get the PDF $q_i$ and model them with piecewise linear functions which have $K$ bins and $K+1$ vertices (bin edges) each. The parameters of these functions can be stored in two matrices: The $|B| \times (K+1)$ matrix $V$ contains the height (vertical coordinate) of the functions at each vertex and the $|B| \times K$ matrix $W$ contains the bin widths (which are adaptive).

A NN outputs the unnormalised matrices $\hat{V}$ and $\hat{W}$. The bin widths should sum to 1, so we normalise the rows of the matrix $\hat{W}$ using the softmax function $\sigma$ and define

$$W_i = \sigma(\hat{W}_i). \quad (19)$$

We want the piecewise linear function $q_i$ to be a PDF, and therefore normalise the rows of $\hat{V}$ according to:

$$V_{i,j} = \frac{\exp(\hat{V}_{i,j})}{\sum_{k=1}^{K} \frac{1}{2}(\exp(\hat{V}_{i,k}) + \exp(\hat{V}_{i,k+1}))W_{i,k}} . \quad (20)$$

Finally, we use linear interpolation to define our PDFs as

$$q_i(x_i^B) = V_{i,b} + \alpha(V_{i,b+1} - V_{i,b}), \qquad (21)$$

where $b$ is the bin that contains the point $x_i^B$ and $\alpha = (x_i^B - \sum_{k=1}^{b-1} W_{i,k})/W_{i,b}$ is the relative position within that bin. By integration we get the piecewise quadratic coupling transform:

$$C_i(x_i^B) = \int_0^{x_i^B} q_i(t)\mathrm{d}t = \frac{\alpha^2}{2}(V_{i,b+1} - V_{i,b})W_{i,b} + \alpha V_{i,b}W_{i,b} + \sum_{k=1}^{b-1} \frac{V_{i,k} + V_{i,k+1}}{2} W_{i,k}. \qquad (22)$$

The corresponding Jacobian determinant is given by

$$\det\left(\frac{\partial C(x^B; m(x^A))}{\partial (x^B)^T}\right) = \prod_{i=1}^{|B|} q_i(x_i^B). \qquad (23)$$

### 3.3 Importance Sampling with Coupling Layers

Having defined the coupling layers, their application for importance sampling is straightforward as they can be used in the same way VEGAS is already applied in existing event generators. The algorithm for a single phase space map, i.e. channel, proceeds as follows.

For each event, we generate a suitable number of uniformly distributed random numbers $x \in \Omega$. These get mapped to non-uniform numbers $y \in \Omega$ using a layered mapping consisting of several coupling layers, as described above. These numbers then serve as input variables for a channel mapping that generates a point $z$ in the target domain. The weight $w$ associated with an event depends on the value of the target function and the Jacobians involved, namely the ones from the coupling layers and the channel mapping itself:

$$w = \left|\det\left(\frac{\partial y(x)}{\partial x^T}\right)\right|\left|\det\left(\frac{\partial z(y)}{\partial y^T}\right)\right| f(z). \qquad (24)$$

Note that we do not use the NN model to generate points in the target domain directly as this could be highly inefficient. For example, if we wanted to generate four-momenta the NN would have to learn four-momentum conservation and on-shell conditions exactly. Using a channel mapping we can implement four-momentum conservation and mass shell conditions directly, lowering the dimensionality of the problem significantly, and also map out known peak structures that might be difficult to infer otherwise.

As for VEGAS we need a mechanism to train our model in order to actually improve the efficiency of the sampler. For this purpose, we define a loss function which gets minimised iteratively using gradient descent. As a loss function we use the Pearson $\chi^2$-divergence between the target function $f$ and the sample distribution $g$ in a minibatch that consists of $n$ sampling points:

$$D_{\chi^2} = \frac{1}{n}\sum_{i=1}^{n} \frac{(f(z_i) - g(z_i))^2}{g(z_i)}, \qquad (25)$$

with points $z_i$ in the target domain, generated from a uniform distribution and transformed by a channel mapping.

Minimising $D_{\chi^2}$ will minimise the variance of a Monte Carlo estimator, as recognised in [33]. Empirically we find that for our applications the mean squared error distance performs better in terms of variance reduction and unweighting-efficiency increase than the Kullback–Leibler divergence.

Our method can be used in a multi-channel approach in the same way as described for VEGAS in Sec. 2.2. It has the additional advantage that we are able to train all mappings for the different channels simultaneously. The channel mappings are aware of each other and do not try to adapt to the same features.

# 4 Results

We have implemented the NN architecture described in the previous section using TENSORFLOW [46]. The NN training is guided using the ADAM optimiser [43]. The default learning rate we use is $10^{-4}$, and gradients calculated for the training are clipped at a value of 100 to avoid instabilities in the training.

We apply our NN-assisted sampling to three standard applications in high-energy physics: the three-body decay of a top quark which features a single importance sampling channel modelling the Breit–Wigner distribution of the intermediate W boson; top-quark pair production in $e^+e^-$ annihilation with the subsequent decay of both top quarks (which also can be modelled by a single importance sampling channel by using the same mapping for both decays); and finally QCD multi-gluon production, with two gluons in the initial state colliding at a fixed centre-of-mass energy and 3 or 4 final-state gluons. For the latter, a multi-channel algorithm as described in the previous section is used, with one NN per independent channel.[3] The required multi-gluon tree-level matrix elements are obtained from SHERPA through its dedicated PYTHON interface [47].

In all cases we compare the performance of the novel NN-assisted importance sampling algorithm with the VEGAS-assisted one which serves as benchmark. We checked that the performance of the VEGAS grids used were not limited by the number of bins or by the number of optimisation steps used.

## 4.1 Top quarks

Top quarks decay predominantly to a W boson and a bottom quark. In turn, the W decays either leptonically or hadronically. This induces an $s$-channel resonance for the W propagator, which is usually described in a phase space sampler by a strongly-peaked Breit–Wigner channel, i.e.

$$g(u) = \frac{1}{\left(s(u) - M_W^2\right)^2 + M_W^2 \Gamma_W^2}, \text{ with } s(u) = M_W \Gamma_W \tan(u) + M_W^2. \tag{26}$$

This channel captures the behaviour of the denominator of the corresponding squared matrix element, but assumes a constant numerator, which renders the channel imperfect for the actual integrand. In the following we study for single top-quark decays and top-quark pair production with subsequent decays how our NN optimisation compares with VEGAS optimisation to remedy such imperfections.

The NN architecture for both top-quark examples consists of 6 piecewise-quadratic coupling layers and 150 bins. The trainings conclude after 6000 optimisation steps, where each step uses a minibatch of 200 phase space points to guide the optimisation.

**Top-quark decays:**  We simulate the decay sequence of a top quark, i.e. $t \to W^+b \to e^+\nu_e b$. With three on-shell final-state particles we have 5 dimensions for the kinematics (the top quark is considered at rest and on-shell). However, we integrate out all dimensions except for the invariant mass of the W-boson decay products and the angle between them. The number of phase space dimensions is therefore $d = 2$. The $s$-channel propagator of the W boson is modelled by a Breit–Wigner distribution in the importance sampling, reducing the variance caused by sampling the strongly-peaked invariant-mass distribution of the lepton-neutrino pair.

The results of a run with $N = 10^6$ events are compared in Tab. 1 with an unoptimised ("Uniform") sampling and a VEGAS-optimised sampling. The Monte Carlo integration result,

---

[3]Here and in the following, "one NN" refers to a connected set of coupling layers, not to the "sub-NN" used within each single coupling layer.

Table 1: Results for sampling the partial top-quark decay width and the total cross section of top-pair production, i.e. the Monte Carlo integral $E_N$ is an estimator for $\Gamma_{t \to b e^+ \nu_e}$ and $\sigma$ for $e^+ e^- \to \gamma \to t[b e^+ \nu_e]\bar{t}[\bar{b} e^- \bar{\nu}_e]$ at $\sqrt{s} = 500\,\text{GeV}$, respectively. Besides $E_N$ and its MC error, we also show the unweighting efficiency $\epsilon_{\text{uw}}$ of the sample, comparing VEGAS optimisation, NN-based optimisation and an unoptimised ("Uniform") distribution. All samples consist of $N = 10^6$ (weighted) points.

| Sample | top decays | | top-pair production | |
|---|---|---|---|---|
| | $\epsilon_{\text{uw}}$ | $E_N$ [GeV] | $\epsilon_{\text{uw}}$ | $E_N$ [fb] |
| Uniform | 59 % | 0.1679(2) | 35 % | 1.5254(8) |
| VEGAS | 50 % | 0.16782(4) | 40 % | 1.5251(1) |
| NN | 84 % | 0.167865(5) | 78 % | 1.52531(2) |

i.e. the partial decay width $E_N$ given by the estimator for $\Gamma_{t \to b e^+ \nu_e}$, is given as a consistency check and to compare its statistical deviation when generating the same number of points $N$ with the alternative sampling methods. The standard deviation obtained with VEGAS is 5 times smaller than for the Uniform sample. Improving on that, the NN sampling has a standard deviation which is 8 times smaller than the VEGAS one. As another figure of merit the unweighting efficiencies $\epsilon_{\text{uw}}$ are compared. Again, the NN has the best (i.e. largest) efficiency, with a value of 0.84 compared to 0.50 for the VEGAS and 0.59 for the Uniform sampling. So while for VEGAS the integral variance is indeed reduced, the unweighting efficiency is somewhat reduced in comparison to the Uniform sampling. This originates from rare outliers in the event weight distribution.

This is illustrated in Fig. 2a, where the distributions of event weights, cf. Eq. (24), for the three samples are shown. The optimal sampler would result in events with identical weights, what leads to a vanishing variance of the integral estimate and an unweighting efficiency of one, cf. Eq. (4). The NN sample features the sharpest peak here and a steeply falling tail towards larger weights, which corresponds to the significantly improved unweighting efficiency. Although the VEGAS sample is also more peaked than the Uniform one, it features large-weight outliers causing the reduced unweighting efficiency.

**Leptonic top-quark pair production:** As a second application, we study the leptonic production of a top–anti-top pair via a virtual photon, and their subsequent leptonic decay, i.e. $e^+ e^- \to \gamma \to t[b e^+ \nu_e]\bar{t}[\bar{b} e^- \bar{\nu}_e]$, at $\sqrt{s} = 500\,\text{GeV}$. This gives us effectively two copies of the top-quark decay chain considered in the previous example, plus the scattering angle between the incoming lepton and the outgoing top quark. This yields a phase space dimensionality of $d = 5$.

Both $s$-channel propagators of the W bosons are modelled by Breit–Wigner distributions, using a single importance sampling channel. Again, a NN sample with $N = 10^6$ points is generated. It is compared in Tab. 1 with an unoptimised and a VEGAS sample of same sizes. The standard deviation of the VEGAS sample is 8 times smaller than the one of the unoptimised sample. The NN sample has the smallest standard deviation, being yet 5 times smaller than the one of the VEGAS sample. The unoptimised and the VEGAS sample have a similar unweighting efficiency of 35 % and 40 %, respectively. The NN one's is about two times better, at 78 %.

Figure 2b depicts the event weight distributions of the three samples. As for the top-decay samples, the NN-optimised sample for top–anti-top production is most strongly peaked, which is in accordance with the small standard deviation and the good unweighting efficiency. The other two samples are significantly broader and have long tails towards large weights.

Overall, the results for top decays and top–anti-top production are similar, which is ex-

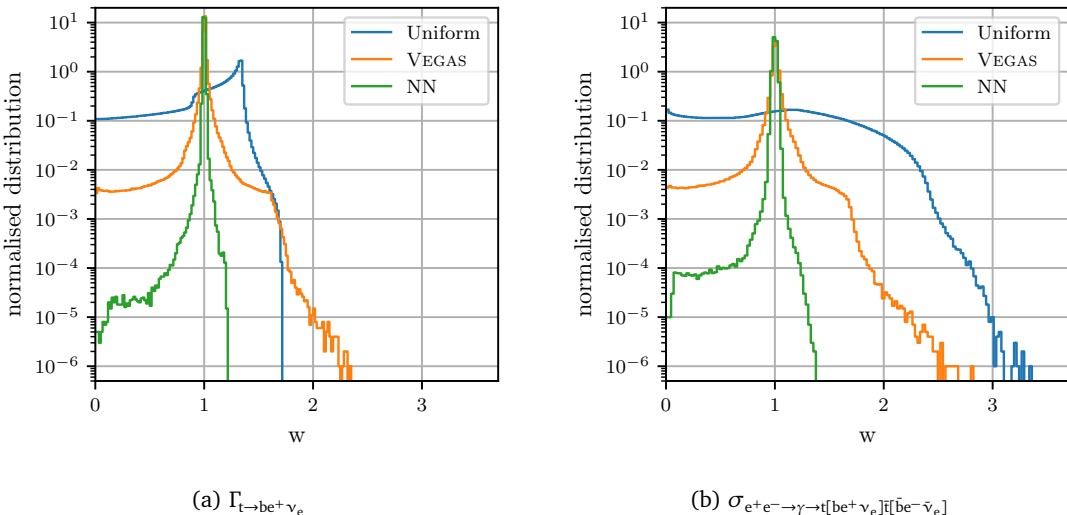

(a) $\Gamma_{t \to b e^+ \nu_e}$  (b) $\sigma_{e^+ e^- \to \gamma \to t[b e^+ \nu_e]\bar{t}[\bar{b} e^- \bar{\nu}_e]}$

Figure 2: Event weight distributions for sampling the partial decay width $\Gamma_{t \to b e^+ \nu_e}$ and the total cross section $\sigma$ for $e^+ e^- \to \gamma \to t[b e^+ \nu_e]\bar{t}[\bar{b} e^- \bar{\nu}_e]$ at $\sqrt{s} = 500\,\text{GeV}$, each with $N = 10^6$ points, comparing VEGAS optimisation, NN-based optimisation and an unoptimised ("Uniform") distribution.

pected because the main difference is that the Breit–Wigner peak appears in one additional dimension for the top–anti-top production, with all other dimensions in phase space not featuring any (strongly) peaked structures. Hence we see a similar shape in the weight distributions, only the unoptimised sample is significantly broader now due to yet another peak it can not adapt to. Compared to the single top decay setup, there is a moderate degradation of the Monte Carlo integration/sampling. The unweighting efficiency is reduced by 7 (20) % for the NN (VEGAS) samples. The unoptimised sample's efficiency is reduced by 40 %.

Finally, we want to study for the case of top–anti-top production how the overall reduction in the width of the weight distributions shown in Fig. 2b translates to more differential observables. We show in Fig. 3 the differential cross section for two observables, the invariant mass of the electron-positron pair $m_{ee}$ and the angle between the electron and the anti-bottom quark $\theta_{e^- \bar{b}}$. Note that the invariant mass $m_{ee}$ depends on the lepton momenta of both top-quark decay sequences, whereas the angle $\theta_{e^- \bar{b}}$ is an observable that depends on the momenta of only the anti-top quark decay sequence. Comparing the results for VEGAS and NN optimisation (again using the samples with equal sizes, $N = 10^6$), we find that both distributions agree and feature nearly equal MC errors across the whole range of the observable. However, the two samples behave differently when we consider the mean weights per bin in the lower panels. With the weights given by the ratio between the integrand and the sampling distribution, cf. Eq. (8), the plots illustrate how close the sampling distribution approximates the actual target. In the perfect case a constant line at 1 would be seen. Any distortion away from 1 directly translates into a broader global weight distribution. For $m_{ee}$, we find that VEGAS samples both tails too often to the expense of the intermediate region between 100 and 250 GeV, whereas the NN sample is nearly constant in comparison. Both samples feature distortions for low $\theta_{e^- \bar{b}}$, although in different directions. As for both VEGAS and the NN most of the weights are very close to 1, which is also reflected in the weight distribution shown in Fig. 2b, the distortions only have a minor impact on the relative MC errors shown in the middle panels.

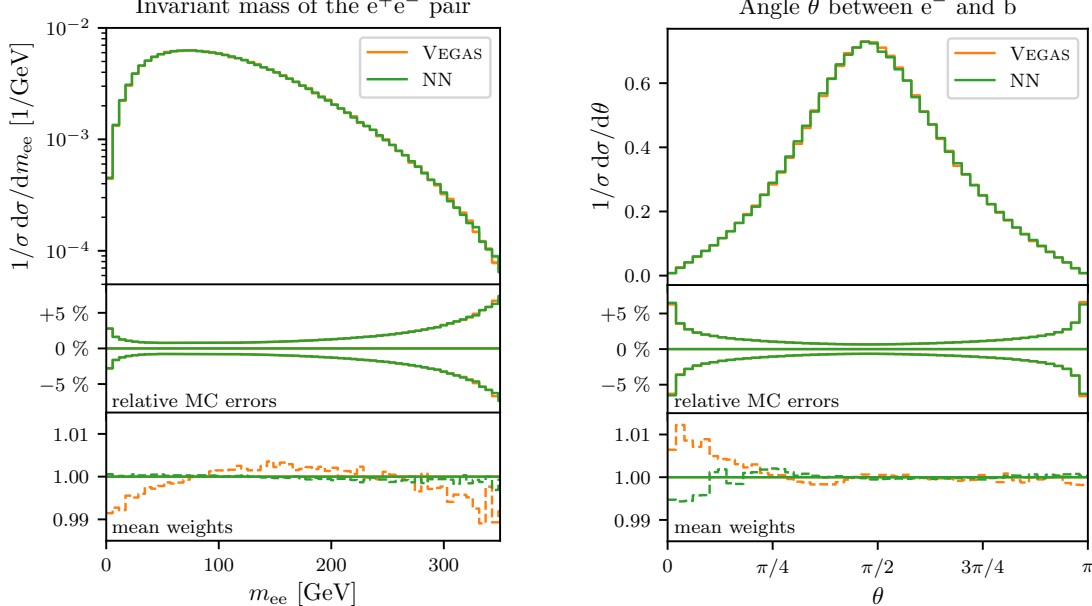

Figure 3: The invariant mass of the electron-positron pair (left) and the angle between the electron and the anti-bottom quark in top-pair production (right). For each observable, we compare the nominal distributions for a VEGAS-optimised and a NN-optimised phase space sampling (upper panes). In the middle panes, Monte Carlo errors for both samples are compared. The lower panes show the mean event weights per bin, highlighting regions of the observables where the sampling distributions over- or undershoot the target.

## 4.2 Gluon-induced multi-jet production

Finally, we test our approach for $gg \to n$ gluons with $n = 3$ and $n = 4$, at a fixed centre-of-mass energy $\sqrt{s} = 1\,\text{TeV}$. For this application, the basic importance sampling density follows the QCD antenna radiation pattern realised by the HAAG algorithm [48,49], with a number of channels that depends on the number of final-state particles. For $n = 3$, HAAG constructs 24 channels, but after mapping channels that differ in the permutation of the momenta only, this boils down to 2 independent channels. For $n = 4$, there are 120 HAAG channels that can be mapped onto 3 independent channels. Therefore, in contrast to the top-quark applications, a multi-channel algorithm is employed, with one independent NN (or VEGAS) per channel. During the training, the NN are all optimised simultaneously, cf. Sec. 3.3.

Another difference with respect to the top-quark examples is the presence of phase space cuts, used to regularise the $n$-gluon cross sections. Hence, the optimisation has to deal with "dead" regions in phase space and therefore with non-continuous integrands.

For regularisation, HAAG uses a cut-off parameter which we set to $s_0 = 900\,\text{GeV}^2$. On the final state we employ a cut on the invariant masses of all parton pairs, i.e. $m_{ij} > 30\,\text{GeV}$, and on the transverse momenta of all particles, $p_{\perp,i} > 30\,\text{GeV}$. To select the jets, we use the anti-$k_t$ algorithm [50] with $R = 0.4$. The renormalisation scale is given by $\mu_R = \sqrt{s}$. Each NN consists of 5 coupling layers and 32 bins. The trainings conclude after a maximum of $10^4$ optimisation steps, where at each step we train the NN on a minibatch of at least 2048 non-zero phase space points.

In Tab. 2, we show the results of sampling the cross section without optimisation ("Uniform"), with VEGAS optimisation and with our NN optimisation. The unweighting efficiencies

Table 2: Results for sampling the total cross section for gluonic jet production at $\sqrt{s} = 1\,\text{TeV}$, i.e. the Monte Carlo integral $E_N$ is an estimator for $\sigma_{gg \to n\,\text{jets}}$. Besides $E_N$ and its MC error, we also show the unweighting efficiency $\epsilon_{\text{uw}}$ of the sample, comparing VEGAS optimisation, NN-based optimisation and an unoptimised ("Uniform") distribution. All samples consist of $N = 10^6$ points with non-zero weights. The table also lists the acceptance rate $P_{\text{acc}}$.

| Sample | 3 jets | | | 4 jets | | |
|---|---|---|---|---|---|---|
| | $\epsilon_{\text{uw}}$ | $E_N$ [pb] | $P_{\text{acc}}$ | $\epsilon_{\text{uw}}$ | $E_N$ [pb] | $P_{\text{acc}}$ |
| Uniform | 3.0 % | 24806(55) | 89 % | 2.7 % | 9869(20) | 57 % |
| VEGAS | 27.7 % | 24813(23) | 32 % | 31.8 % | 9868(10) | 17 % |
| NN | 64.3 % | 24847(21) | 34 % | 33.6 % | 9859(10) | 16 % |

$\epsilon_{\text{uw}}$ for $n = 3, 4$ are about 3 % for the unoptimised sampling, and increase to about 30 % by VEGAS optimisation. The NN optimisation achieves to surpass VEGAS for $n = 3$ by a factor of two, whereas for $n = 4$ we find no significant improvement over VEGAS. Both VEGAS and NN optimisation gives similar improvements for the estimate of the standard deviation for $n = 3$ and $n = 4$. We also quote in Tab. 2 the acceptance rate $P_{\text{acc}} = N/N_{\text{trials}}$, i.e. the probability that a proposed point passes the phase space cuts and hence provides a finite contribution to the integral result. In our gluon production setup, the cuts regularise the matrix elements, and therefore the matrix element value is expected to be larger close to these cuts than elsewhere. It is therefore unsurprising that both VEGAS and NN optimisation lead to a decrease in $P_{\text{acc}}$, as they enhance the sampling rate close to the cuts, with the side effect of proposing points also outside of the cuts (since the bin edges of both methods will not perfectly coincide with the cuts).

The event weight distributions for the samples are compared with each other in Fig. 4. For 3-jet production, we find that the NN optimisation gives the most strongly peaked weight distribution. The situation is more ambiguous for 4-jet production. Both the VEGAS and NN optimisation significantly sharpen the weight distribution, in fact providing quite similar outcomes. However, while the NN optimisation results in a slightly more pronounced peak compared to VEGAS and a slightly faster fall-off towards large weights, it depletes less quickly towards small weights. In particular for the 3-jet case it might be surprising that we find a comparable estimate for the standard deviation for NN and VEGAS optimisation, although the weight distribution is narrower in the NN case. This apparent discrepancy originates from the higher fraction of zero-weight events for the optimised samples, i.e. events that fall outside the physical phase space volume and are thus not accepted. The standard deviation of the integral estimate is in such a case largely determined by the corresponding acceptance rate, since the weight distribution will then actually contain two peaks: the one at a finite value and one at $w = 0$. A further improvement in the sampling accuracy would therefore require a modification of the optimisation to reduce the number of discarded phase space points. The unweighting efficiency is not affected by $P_{\text{acc}} < 1$, since it takes into account non-zero weights only.

In Fig. 5 we depict the transverse momentum distributions for the jet with the smallest transverse momentum $p_\perp$ in three- and four-jet production, i.e. the third and the fourth jet, respectively, again comparing the NN-optimised sample with a VEGAS-optimised one. In the comparisons of the mean weight per bin distributions (lower panels) we find a different behaviour for the two optimisation methods. For three-jet production, Fig. 5a, the NN weights stay very close to one for $p_T \lesssim 240\,\text{GeV}$, whereas VEGAS samples the lower-most two $p_\perp$ bins with weights smaller than unity, which is compensated by weights larger than unity already

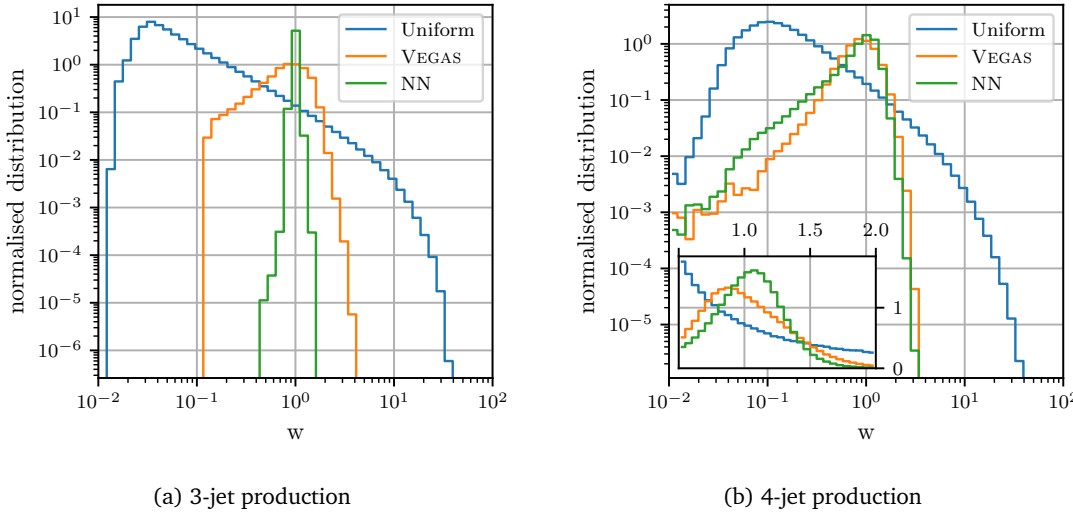

(a) 3-jet production

(b) 4-jet production

Figure 4: Event weight distributions for sampling the total cross section for gg →$n$ jets for $\sqrt{s} = 1\,\text{TeV}$ with $N = 10^6$ points, comparing VEGAS optimisation, NN-based optimisation and an unoptimised ("Uniform") distribution. Note that we now use a logarithmic scale for the $x$ axis. The inset plot in (b) shows the peak region in more detail and using a linear scale.

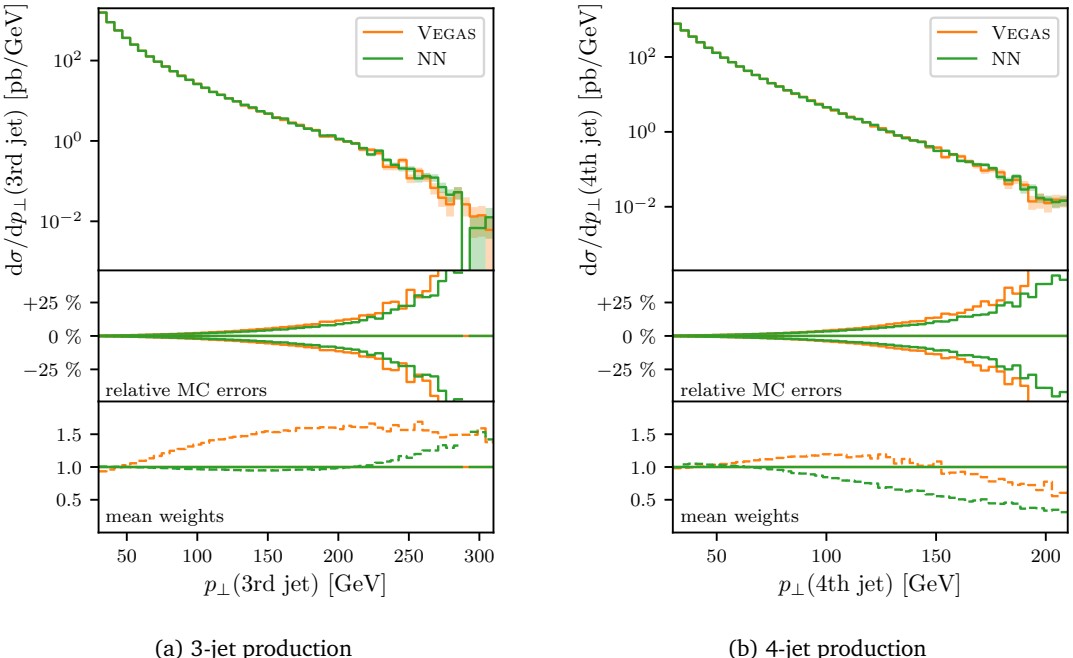

(a) 3-jet production

(b) 4-jet production

Figure 5: The transverse momentum of the smallest-$p_\perp$ jet in three-gluon production (left) and in four-gluon production (right). For both observables, we compare the nominal distributions for a VEGAS-optimised and a NN-optimised phase space sampling (upper panes). In the middle panes, Monte Carlo errors for both samples are compared. The lower panes show the mean event weights per bin.

above 50 GeV. For four-jet production, Fig. 5b, the NN sample differs from unity for $p_\perp$ values larger than 80 GeV. Then the weights become increasingly smaller, which corresponds to the long tail of the weight distribution towards smaller weights in Fig. 4b. For VEGAS, weights again begin to differ from unity above 50 GeV. However, there is a turning point and therefore the weights remain closer to unity compared to the ones of the NN sample above 100 GeV. Hence, judging the sample quality is less straightforward in the four-jet case, whereas the NN sample is clearly better in the three-jet case. This is in agreement with the very similar global sample performance given in Tab. 2.

Considering the relative MC errors in the middle panel of Fig. 5b we observe that although the NN distribution differs from the integrand more than the VEGAS distribution in the high-$p_\perp$ bins, it still leads to smaller relative errors. This is, however, just a consequence of the statistics: the NN sample features smaller weights in this region as it oversamples the target. Therefore, it generates more events per bin than VEGAS which results in a smaller variance.

## 5  Conclusions

We have conducted a proof-of-principle study for applying Neural Importance Sampling with piecewise-quadratic coupling layers to optimise phase space sampling in Monte Carlo integration problems in high-energy physics. The approach fulfils the requirements needed to guarantee a faithful sampling of the target distribution. In particular, full phase space coverage is guaranteed. We have investigated the performance of the approach by employing it as a drop-in replacement of the widely used VEGAS optimiser, which we use for comparison benchmarks. Specifically, we have studied the efficiency of the approach both for the integration result and for the generation of weighted and unweighted event samples for the decay width of a top quark, for the cross section of leptonic production of a top-quark pair with subsequent decays; and for the cross sections of gluonic 3-jet and 4-jet production.

We find a significantly improved sampling performance for the simpler examples with a phase space dimensionality up to $d = 5$, namely top decays, top pair production and 3-jet production. For the more complex example of 4-jet production with $d = 8$ and an increased number of importance sampling channels, we have not been able to outperform VEGAS, e.g. the gain factor in the unweighting efficiency dropped from 2.3 for 3-jet production to 1.1 for 4-jet production. Since the complexity of the NN architecture and the number of events per training batch was limited by our computing resources, we expect that the result for the 4-jet case can be improved by using more powerful hardware and/or optimising the implementation. Though, even then the computational challenge would emerge again for 5-jet production, and it is left to further studies to improve the scaling behaviour of the ansatz. Our findings are consistent with those in another study [36], where increasing the final-state multiplicity (and hence the number of channels) in V + jets production also leads to a rapid reduction in the gain factor.

However, the results for the top quarks and the 3-jet production are promising and indicate that conventional optimisers such as VEGAS can potentially be outperformed by NN-based approaches also for more complex problems in the future. To this end the computational challenges outlined above need to be addressed. In future research we will therefore aim to extend the range in final-state multiplicity while keeping the training costs at an acceptable level, and—if successful—to implement the new sampling techniques within the SHERPA general-purpose event generator framework. A starting point should be the further study and comparison of alternative ways to integrate our NN approach within multi-channel sampling, beginning with our ansatz and the one proposed in [36], to find out if the scaling behaviour can be optimised. On the purely NN side, the exploration of possible extensions or alternatives to piecewise-quadratic coupling layers is promising, such as [51]. Also adversarial training has

the potential to reduce training times significantly. The limitation of the statistical accuracy by a large number of zero-weight events found in the jet-production examples furthermore suggests that it is worthwhile to investigate the construction of optimised importance sampling maps that better respect common phase space cuts, or alternatively to modify the optimisation procedure to further reduce the generation of points outside the fiducial phase space volume.

## Acknowledgements

We are grateful to Stefan Höche for fruitful discussion. We would also like to thank Tilman Plehn, Anja Butter and Ramon Winterhalder for useful discussions.

This work has received funding from the European Union's Horizon 2020 research and innovation programme as part of the Marie Skłodowska-Curie Innovative Training Network MCnetITN3 (grant agreement no. 722104). SS acknowledges support through the Fulbright–Cottrell Award and from BMBF (contract 05H18MGCA1).

# A  Auxiliary jet $p_\perp$ distributions in multi-jet production

In this appendix we compile additional plots of the jet $p_\perp$ distributions in 3- and 4-gluon production from gluon annihilation at $\sqrt{s} = 1\,\mathrm{TeV}$. Details on the calculational setup are given in Sec. 4.2.

The leading and second-leading jet $p_\perp$ distribution in 3-gluon production are depicted in Fig. 6a. The leading, second- and third-leading jet $p_\perp$ distributions in 4-gluon production are shown in Fig. 6b.

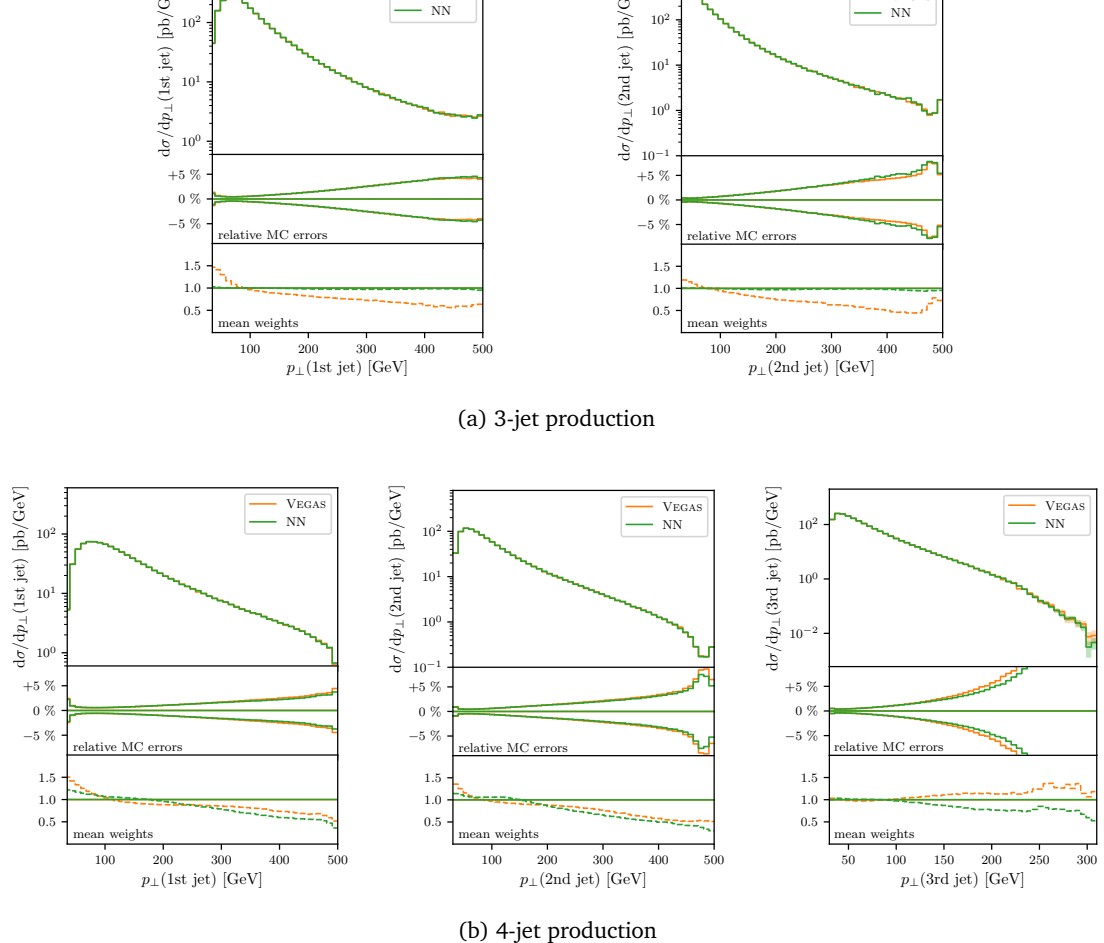

(a) 3-jet production

(b) 4-jet production

Figure 6: Distributions of the transverse momentum $p_\perp$ of the leading and second-leading jets in three-gluon production **(a)** and for the leading, second- and third-leading jets in four-gluon production **(b)**. For each observable, we compare the nominal distributions for a VEGAS-optimised and a NN-optimised phase space sampling (upper panes). In the middle panes of each plot, Monte Carlo errors for both samples are compared. The lower panes show the mean event weights per bin.

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
