# Peer review of "Exploring phase space with Neural Importance Sampling"

_SciPost Physics, doi:SciPost Phys. 8, 069 (2020)_

## Round 2 · Referee Report · Anonymous (Referee 1) · 2020-2-26

Strengths

1 - The paper provides a thorough review of existing methods for adaptive Monte-Carlo integration. It outlines clearly how known techniques based on Neural Networks can fail to fill the full phase space and therefore yield a biased integration result.
2 - The authors construct a novel adaptive integration algorithm based on Neural Networks and Normalizing Flows. They apply this algorithm to various test cases, including a three-body decay, top-quark pair production and decay at a lepton collider, and partonic three- and four-jet production.
3 - They present a comprehensive comparison between one of the best existing adaptive integrators (Vegas) and their new technique for weight distributions, event generation efficiencies and integration uncertainties on physical observables of potential experimental interest.

Report

The paper provides a thorough review of existing methods for adaptive Monte-Carlo integration. It outlines clearly how some techniques based on Neural Networks can fail to fill the full phase space and therefore yield a biased integration result.
The authors then proceed to construct a novel adaptive integration algorithm based on Neural Networks and Normalizing Flows. They apply this algorithm to various test cases, including a three-body decay, top-quark pair production and decay at a lepton collider, and partonic three- and four-jet production.
They present a comprehensive comparison between one of the best existing adaptive integrators (Vegas) and their new technique for weight distributions, event generation efficiencies and integration uncertainties on physical observables of potential experimental interest. They conclude that the efficiency of their novel integrator is superior to Vegas at low final-state multiplicity, but drops at higher multiplicity due to a lower compute efficiency.

I highly recommend this preprint for publication. I would suggest a minor modification, which is to include a reference to Nucl.Phys. B9 (1969) 568-576 in the last paragraph of Sec.2, where channel construction is discussed.
  • validity: -
  • significance: -
  • originality: -
  • clarity: -
  • formatting: -
  • grammar: -

Author:  Steffen Schumann  on 2020-02-27  [id 746]

(in reply to Report 1 on 2020-02-26)
Category:
remark

We would like to thank the reviewer for the comments. We will include a reference to the suggested article in the revised version of our manuscript.

---

## Round 2 · Referee Report · Tilman Plehn (Referee 2) · 2020-3-2

Strengths

The paper is one of the first application of machine learning to MC generation and one of the first applications of normalizing flow networks. It is very well done, technically state of the art, and indicates serious potential.

Weaknesses

Just little stuff, plus why does the bloody 4-jet case not really work???

Report

I really like the paper, it should be published with minor changes. Requested changes are largely suggestions concerning the presentation.

Requested changes

1- conceptually, I am not sure I understand the two figures of merit and how they are linked. Any chance you could explain more about this on p.3, where they are introduced? 2- I am sorry, but I find the beginning of Sec.2.3 hard to read for NN-non-experts. Any chance you could illustrate it better. Do you have an image illustrating your network structure? 3- I am not super-happy with the coverage arguments, which are partly linked to mapping an infinite weight space on a finite phase space. What exactly is not possible there? We did look into this problem in our Bayesian classification paper, and it is painful with the sigmoid function, but I do not understand the fundamental issues you are implying; 4- looking at Fig.1 (lower) I would argue that all you see is a lack of training data in the tails, as we know causes problems for GANs universally, for instance in event generation or in unfolding. Why do you consider this a fundamental problem? 5- concerning top decays - why is VEGAS not good? Or is flat just unusually good, because you mapped out the BW? 6- please explain Fig.2 more carefully, like what does it show, and what do the peaks imply? By the way, why do not use a linear weight scale for Fig.2? I seems not obvious given the scale-less latent space; 7- at the latest towards the bottom of p.11 I am wondering if the definition of the loss function is related to the two figures of merit, please say something about this; 8- in Sec. 4.2 the elephant sitting in the room is the lack of improvement in the 4-jet case. Please say more about this, including maybe what the reason might be and what you tried to improve it. Do you really think it can be improved with better GPUs? It looks like a problem of multi-channel naively combined with NNs, also after reading Ho-Chi's paper; 9- On with some comments concerning the presentation: I have to admit that I do not like switching back between integral representations like Eq.(1) and sum representations like Eq.(2). Why do you do that? In any case, if you do this please make sure everything is well defined... 10- please explain the phase space coverage for non-experts, because it is the main constraint on applying NNs to phase space generation, c.f. p.3, point (i); 11- at the beginning of Sec.2.1 it might be useful to mention that all of this is just a variable transform where we only really case about the Jacobian? 12- please explain Eq.(11) more carefully, and how it relates to Eq.(7); 13- what does the \sim before Eq.(12) mean? 14- the symbols in Eq.(23) are not clearly defined there, I think; 15- why the funny E_N notation in Tab.1? 16- moving Eq.(24) to somewhere in the introduction might make the introduction a little less dry/formal; 17- I am sorry, but the argument in the second paragraph on p.13 (The event weight distribution) is not clear to me. Can you have a look at this paragraph, please? 18- for applications of flow networks in physics it would be nice to cite the paper(s) by Ulli Koethe. They are not particle physics, but they are the first physics applications I know of.

  • validity: top
  • significance: top
  • originality: top
  • clarity: high
  • formatting: perfect
  • grammar: perfect

Author:  Steffen Schumann  on 2020-03-27  [id 779]

(in reply to Report 2 by Tilman Plehn on 2020-03-02)
Category:
answer to question
correction

Dear Tilman,

thank you for the detailed comments and suggestions. We have tried to answer all your queries and give our detailed replies below. An updated version (v3) has been submitted to the arXiv as well as SciPost.

We hope the the revised version can be considered for publication in SciPost.

All the best,

Steffen (for the authors)

R1 -- We have somewhat extended the discussion of the unweighting efficiency and the variance and mention their complementarity for optimising a sampler.

R2 -- We added a more detailed introduction to Sec. 2.3, this was indeed too brief before.

R3/4/10 -- We have extended the discussion and added additional data to Fig. 1a, illustrating the fact
that even a significant increase in training data does not yield a satisfactory phase space coverage.
In consequence the non-surjective NN sampler does not reproduce the desired target distribution.

R5/6 -- We extended the discussion on the event weight distribution, i.e. Fig. 2. In the discussion of the
top-quark decay example we explicitly refer to the complementarity of improving the variance vs. the unweighting
efficiency.

R7/14 -- We have clarified the relation between the loss function and the figurues of merit, adopting the
notation to enhance clarity. To this end, we have reexpressed the loss function in terms of the variables used
throughout, providing further explanation and a reference.

R8 -- Indeed, no definite answer on why the performance for the higher-multiplicity processes drops can be given yet.
We more clearly stress this in the corresponding paragraph as well as the conclusions now, we further removed the
misleading statement that the number of training epochs was limited by our computational resources.

R9 -- We have carefully checked our notation and would like to stick with both the integral and the sum representation,
with the latter providing the numerical estimate for the former. To somewhat clarify matters we have moved the
reference to the integrand mean to Eq. (2).

R11 -- We have adjusted the first sentence of the second paragraph in Sec. 2.1

R12 -- We reworded the description of Eq. (11) and added a reference to Eq. (7)

R13 -- This was meant to represent that variable x is distributed according to $p_X(x)$, we now explicitly state this in
words.

R15 -- Given that our integrals have different physical meaning, i.e. decay widths or various cross sections, we hope that
using integral estimate $E_N$ helps to relate the physics applications to the discussion of the method in Secs. 2/3.
For clarification we have adjusted the captions in the various results tables.

R16 -- We have added a reference to the physics examples to be discussed to the introduction.

R17 -- We revised the corresponding paragraph.

R18 -- Thanks for the suggestion for additional references. The work of Koethe is certainly interesting and related in
subject. However, given the rather loose connection to physics applications therein, we would in fairness have to include
a long list of other references to flow network applications as well, which we feel is beyond the scope of the paper.

---

## Round 3 · Referee Report · Tilman Plehn · 2020-3-27

Report

Thank you for taking into account my comments. I disagree with the referencing aspect, but then I am also not the author of the paper...

---

## Round 3 · Author Response

With this revision we have tried to address all the comments and suggestions raised by the two referees.
We hope that the paper in its present form can be published in SciPost.

---

## Round 3 · List of Changes

* added reference to Byckling, Kajantie Nucl. Phys. B 9 (1969) 568

* updated references

* introduced variance V_N at first appearance after Eq. (3)
* updated discussion of figures of merit, i.e. V_N and e_UW
* We have inverted the ordering of the sampler criteria (i) and (ii) to better frame the coverage argument, slightly reworded, footnote added
* added reference to the physics examples being presented later on to the introduction
* clarified relation between loss function and figures of merit, thereby adapting the notation for improved clarity
* extended the discussion of the event weight distribution for the top-quark decay example, i.e. Fig. 2.
* reworded description of Eq. (11) a little, in particular adding a reference to Eq. (7)
* updated captions of results tables, making explicit that we show the *MC integral* E_N as
an *estimator* for the physical widths/cross sections
* updated Fig 1a to illustrate that increasing the number of data points in the training has a diminishing
return of investment in terms of asymptotic phase space coverage for the non-surjective NN; also
added some minor clarifications in the main text
* Updated wording in the conclusion in regards of more complex multi-channel and high-multiplicity integration problems

---

## Editorial Decision

published